# Effects of diuretic administration on outcomes of extracorporeal shockwave lithotripsy: A systematic review and meta-analysis

Zhenghao Wang[☯], Yunjin Bai[☯], Jia Wang[iD]*

Department of Urology, Institute of Urology, West China Hospital, Sichuan University, Chengdu, China

☯ These authors contributed equally to this work.
* wangjia201707@163.com

## Abstract

The present systematic review and meta-analysis of randomized controlled trials (RCTs) was conducted for investigating the effect of diuretics on the outcomes of shockwave lithotripsy (SWL) for the treatment of urinary stones. We performed searches of PubMed, Web of science, Embase, EBSCO, and Cochrane library databases from inception to November 2019. RCTs were selected for assessing the effects of diuretics on fragmentation and clearance of urinary stones. The search strategy and study selection process were performed in accordance with the PRISMA guidelines. Four RCTs were included in the meta-analysis. Overall, intervention groups experienced significant improvements in fragmentation compared with the control groups (risk ratio [RR] = 1.14, 95% confidence interval [CI] = 1.05–1.03, P = 0.02). However, stone clearance did not significantly differ between the intervention and control groups (RR = 1.23, 95% CI = 0.97–1.56, P = 0.08). The total numbers of shocks and sessions required were significantly reduced by the use of diuretics. Diuretics significantly enhance stone fragmentation for patients undergoing SWL. However, the improvement in stone clearance appears to be insignificant.

## Introduction

Urinary stone disease is the third-most common disease of the urinary tract worldwide, and it is reported to affect 1%–5% of the population of Asia, 5%–9% of the population of Europe, and 13% of the population of North America. Along with its high prevalence, the condition is also reported to have a high rate of recurrence; 50% within 5–10 years and 75% within 20 years [1]. Patients with kidney stones often suffer from short-term symptoms including acute renal colic pain, nausea, vomiting, and hematuria, as well as long-term complications such as chronic urinary tract obstruction, hydronephrosis, and renal damage [2]. Shockwave lithotripsy (SWL) was originally introduced in clinical practice in 1983, and it has since become widely accepted as the gold standard therapy for kidney stones of under 2 cm of diameter [3]. Compared with endoscopic and open surgical procedures, SWL is a minimally invasive procedure with

**Data Availability Statement:** All data in our study are available from the PubMed, EMbase, Web of science, EBSCO, Cochrane library.

**Funding:** The authors received no specific funding for this work.

**Competing interests:** The authors have declared that no competing interests exist.

reduced requirements for anesthesia; therefore it is associated with high rates of patient acceptance [4]. However, because of the nature of SWL, results are not immediate, and some patients require repeat sessions for removing residual stone fragments, thus increasing the cost and possibility of complications. Moreover, 10%–20% of fragmented stones will grow in size over time [5], resulting in severe pain that can significantly affect the quality of life and vocational responsibilities [6].

Adjuvant interventions are urgently required for improving the results of SWL in terms of residual stone removal and overall efficacy. For reducing the necessity of more invasive treatments such as ureteroscopy, less-invasive interventions including inversion therapy, mechanical percussion, and drug therapy have been explored [7]. Among these, pharmacotherapy is considered as a promising approach, with medicines such as calcium channel blockers, α-adrenergic blockers, nonsteroidal anti-inflammatory drugs, and progesterone being proven to have beneficial effects on the expulsion of stones and efficacy of SWL [8–11]. However, the effects of diuretics on the success of SWL remain unclear from the present literature.

We performed a systematic review and meta-analysis of randomized controlled trials (RCTs) for investigating the effects of diuretic administration during SWL on outcomes. The results of this investigation may guide clinical decision-making for the administration of diuretics in the context of SWL.

## Materials and methods

This systematic review and meta-analysis followed the guidelines of the Preferred Reporting Items for Systematic Reviews and Meta-analysis (PRISMA) statement and the Cochrane Handbook for Systematic Reviews of Interventions [12–13]. Ethical approval and patient consent were not required because all analyses were based on previously published studies.

### Literature search and selection criteria

We systematically searched several databases including PubMed, EMbase, Web of science, EBSCO, and the Cochrane library from inception to November 2019 with the following keywords: "diuretic," "shock wave lithotripsy," "furosemide," "drug therapy," and "urolithiasis." The reference lists of retrieved studies and relevant reviews were hand-searched, and the process mentioned above was repeatedly performed for ensuring that all eligible studies were included.

Inclusion criteria were as follows: (1) RCT study design, (2) the intervention was SWL with the use of diuretics versus SWL with placebo (or with no intervention), (3) adequate reporting of data provided for analysis, and (4) availability of the entire text. Studies reported in all languages were included.

### Data extraction and outcome measures

Baseline information that was extracted from the original studies included the following: first author, published year, number of patients, patient age and gender distributions, description of calculus, and detail methods for the two groups. Data were independently extracted by two investigators (W.Z.H and B.Y.J). Discrepancies were resolved by consensus.

The primary outcomes were stone clearance and fragmentation. Secondary outcomes were the total number of shocks and number of sessions required.

### Quality assessment of individual studies

The methodological quality of each RCT was assessed according to the Jadad Scale, which comprises the following three evaluation elements: randomization (0–2 points), blinding (0–2

points), and dropouts and withdrawals (0–1 points) [14]. One point was awarded for each element that was conducted and appropriately described in the original article. The total score varies from 0 to 5 points. An article with a Jadad score of ≤2 is considered to be of low quality, while a Jadad score of ≥3 indicates the high quality of a study [15]

### Statistical analysis

Risk ratios (RR) with 95% confidence intervals (CIs) were calculated for dichotomous outcomes. Heterogeneity was evaluated using the $I^2$ statistic, with $I^2 > 50\%$ taken to indicate significant heterogeneity [16]. Sensitivity analysis was performed for evaluating the influence of a single study on the overall estimate by omitting one study in turn or performing subgroup analysis. The random-effects model was used for meta-analysis. Owing to the limited number of included studies (<10), publication bias was not assessed. Statistical significance was accepted at P < 0.05. All statistical analyses were performed using Review Manager Software Version 5.3 (The Cochrane Collaboration, Software Update, Oxford, UK).

## Results

### Literature search, study characteristics, and quality assessment

A total of 389 articles were initially identified from database searches. After the removal of duplicates, 255 articles were retained. Of these, 247 were excluded from analysis following the screening of the abstracts and titles, three were excluded due to study design, and one was excluded because of insufficient data. Four RCTs were identified for satisfying the inclusion criteria, and they were finally enrolled in this meta-analysis [17–20]. The article selection process was performed in accordance with the PRISMA guidelines (Fig 1).

Baseline characteristics of the four included RCTs are shown in Table 1. These studies were published between 2001 and 2017, and the total sample size was 349. Patients in three of the studies [17–19] received 40-mg furosemide at the initiation of SWL, while one study [20] describes the administration of 20-mg furosemide at the initiation of SWL. Sabharwal et al. used shocks at a frequency of 80/min starting at 7 kV with dose escalation up to 16 kV until either the stone fragmented or the maximum of 1,500 or 2,000 shocks was reached (per session) for renal or upper ureteric calculi, respectively. Up to three sessions were performed. Zomorrodi et al. described the administration of 3,500 shocks with an energy of 9–13 kV per session in up to three sessions. Azm et al. administered shocks at a rate of 90/min at 10 kV with dose escalation up to 18 kV (in up to four sessions). Lastly, Yoon et al. reported the use of 3,000 shocks in one session. The evaluated of the stone clearance for three studies [17–19] were 3 months and one [20] is for three weeks.

Although all the RCTs reported the rate of stone clearance and fragmentation, only two RCTs [18–19] described the total number of shocks and sessions required.

The Jadad scores of the included studies varied from two to four. One study [18] was considered to be low quality while other three studies [17, 19, 20] were considered to be high quality.

### Primary outcome: stone clearance and fragmentation

A random-effects model was used for analyzing the primary outcomes. Compared with control groups, our results indicated that the use of diuretic significantly improved the fragmentation achieved by SWL (RR = 1.14, 95% CI = 1.05–1.03; P = 0.02) with insignificant heterogeneity among the studies ($I^2$ = 0%, P = 0.42, Fig 2). Although the outcome of stone clearance showed some differences between the studies, this was not noted to be statistically

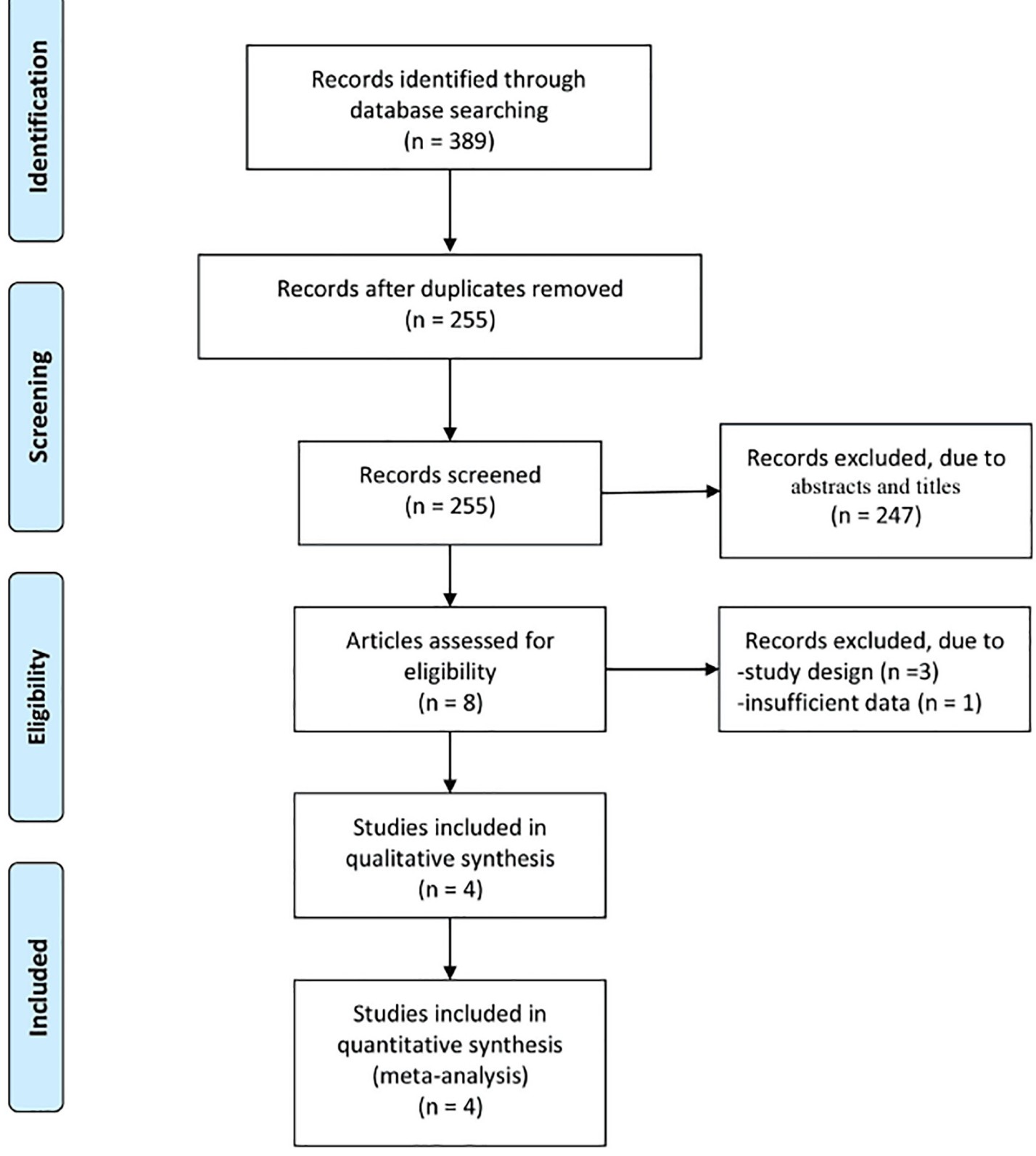

**Fig 1. Flow diagram of study searching and selection process.**

**Table 1. Characteristics of the included studied.**

| No. | Author | year | Experimental group | | | | | Control group | | | | | Jadad score |
|---|---|---|---|---|---|---|---|---|---|---|---|---|---|
| | | | Number (n) | Age (Mean ±SD) | Male (n) | Calculus size (mm) | Method | Number (n) | Age (Mean) | Male (n) | Calculus size (mm) | Method | |
| 1 | Sabharwal | 2017 | 48 | 38.5±10.5 | 31 | 9.4±3.1 | 40 mg furosemide at the start of SWL | 48 | 39.4 ±10.9 | 30 | 9.2±3.1 | SWL with placebo | 4 |
| 2 | Zomorrodi | 2008 | 44 | 12 to 52 | - | 10–18 | 40 mg furosemide at the start of the SWL | 43 | 12 to 52 | - | 8–16 | SWL | 2 |
| 3 | Azm | 2001 | 52 | 38.8±10 | - | 9.3±2.3 | 40 mg furosemide | 54 | 38.8±10 | - | 9.6±2.5 | SWL | 3 |
| 4 | Yoon | 2002 | 30 | 43.1±11.3 | | 8.9±5.1 | 20 mg furosemide at the start of SWL | 30 | 44.5 ±11.3 | | 9.2±4.8 | SWL with placebo | 4 |

significant (RR = 1.23, 95% CI = 0.97–1.56, P = 0.08) with significant heterogeneity ($I^2$ = 74%, P = 0.01, Fig 3).

### Secondary outcomes: total number of shocks and sessions required

The study by Sabharwal et al. reported mean total numbers of shocks in experimental and control groups of 3,661.4 ± 1,946 and 3,894.7 ± 2,254, respectively (P<0.05), and a mean number of sessions of 2.12 ± 1.17 and 2.25 ± 1.3, respectively (P<0.05). Zomorrodi et al. reported 5,300 and 6,293 shocks, respectively (P < 0.05) and 1.5 and 1.92 sessions, respectively (P < 0.05). The above results could not be subjected to meta-analysis due to incomplete data; however, the numbers of shocks and sessions were not significantly affected by the use of diuretic.

### Sensitivity analysis

Significant heterogeneity was observed for the outcome of stone clearance. Sensitivity analysis was performed to evaluate the stability of the results. After removing the study by Yoon et al., heterogeneity was low ($I^2$ = 2%, P = 0.36) and the outcome of stone clearance remained statistically insignificant (RR = 1.11, 95% CI = 1.00–1.23, P = 0.02, Fig 4).

## Discussion

Our results suggested that the use of diuretics during SWL improved the fragmentation of urinary stones. In present clinical practice, SWL is the most frequently used approach for the treatment of urolithiasis [18]. Despite its several advantages, incomplete fragmentation may occur, requiring multiple SWL sessions and resulting in pain caused by large stone fragments. The success of SWL is influenced by several factors, including the stone characteristics (site,

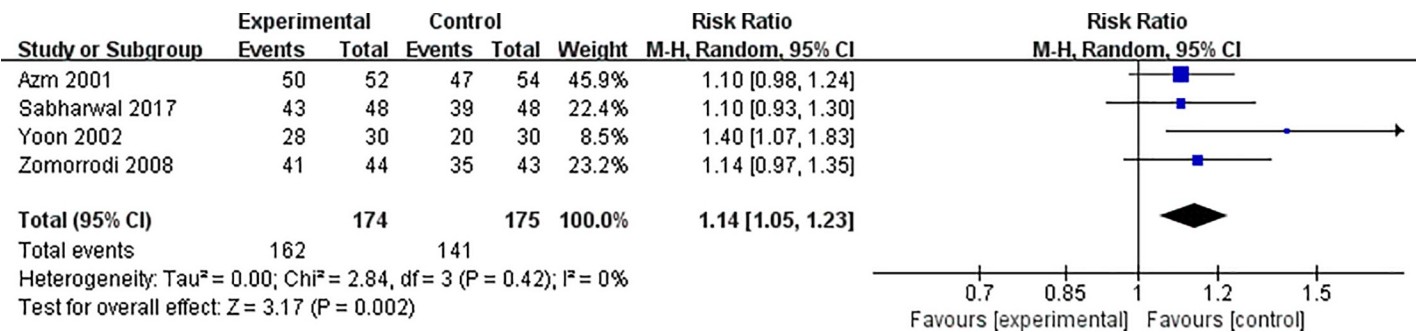

**Fig 2. Forest plot for the meta-analysis of fragmentation.**

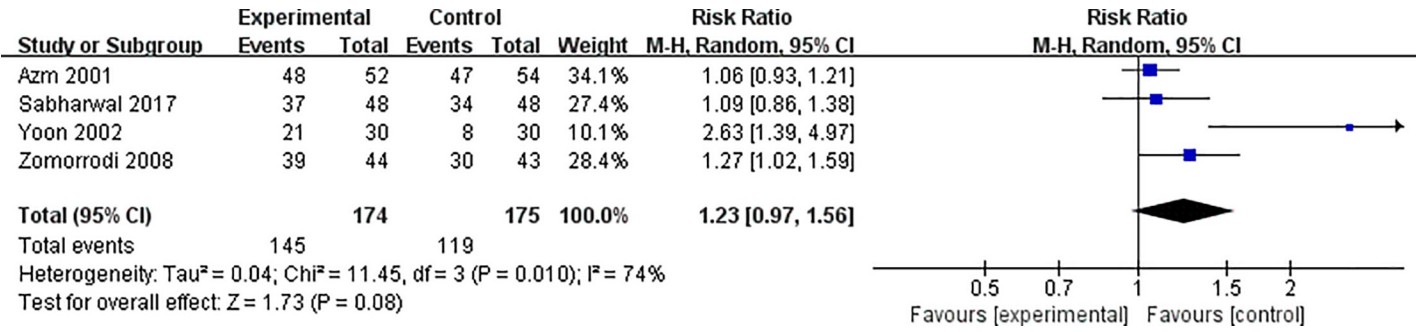

**Fig 3. Forest plot for the meta-analysis of stone clearance.**

burden, and type) and the condition of the kidney (degree of obstruction, renal unit functionality) [21]. Adjuvant therapies can often improve the outcomes of SWL. Ureteral stenting during SWL has been proposed; however, its disadvantages overpower its benefits [22]. Some studies have also reported that diuresis could improve the outcomes of SWL, yet its clinical benefit remains unclear. Herein, we investigated the effects of diuretic administration on the outcomes of SWL.

Diuresis causes the production of a fluid film, which forms a layer on the surface of the stone, possibly contributing to the increase in the success rate [23]. Additionally, after the outer shell of the stones are damaged by the shocks, the permeation rate of urine into the stone increases, improving the effects of subsequent shock waves on the stone core [24]. Diuresis causes the production of urine during SWL, creating a liquid interface on the shell surface and between the damaged shell and core. This hypothesis is supported by the fact that the total number of shocks and sessions required for complete lithotripsy decreased, presumably due to enhanced fragmentation.

The heterogeneity in stone clearance was initially attributed to clinical heterogeneity coming from the study by Yoon et al. In this study, the outcome was assessed three weeks after the treatment, while the other studies reported this parameter three months post-treatment. Furthermore, in Yoon's study, only one SWL session was performed, while the other three studies used a maximum of four sessions. Nevertheless, after the exclusion of this study, the differences in stone clearance remained insignificant. Despite the positive effects of diuretics on fragmentation, stone clearance is influenced by multiple factors. Additionally, multiple SWL sessions may obscure the effects of diuretics on stone clearance. In the Yoon et al. study [20], the patients underwent fewer SWL sessions, and significant differences in stone clearance between patients who received diuretics or placebo were reported. Residual stones were

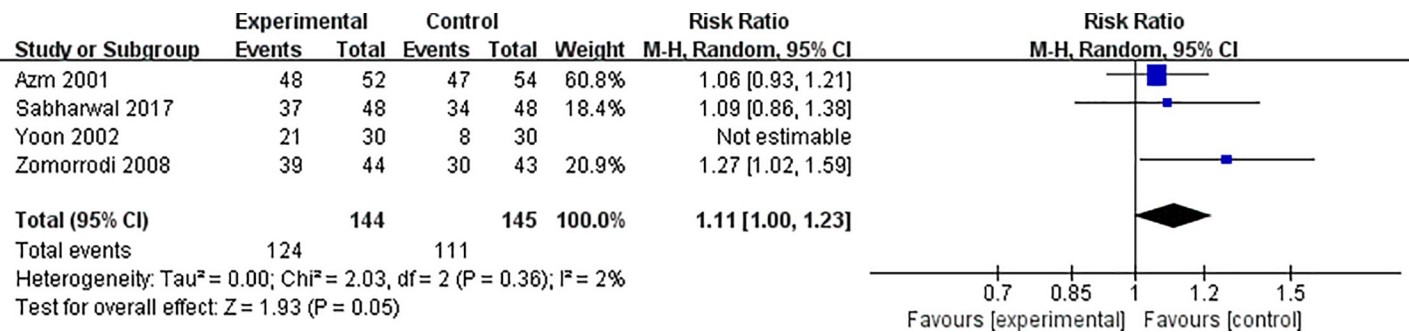

**Fig 4. Forest plot for the meta-analysis of stone clearance after sensitivity analysis.**

evaluated three months post-treatment in all the studies included in our final analysis. Although we did not find the use of diuretics to cause a significant improvement in stone clearance in the long-term following repeated sessions of SWL, whether diuretics reduce the time required for the stone clearance remains unclear.

To the best of our knowledge, this is the first systematic review and meta-analysis investigating the effects of diuretics administration during SWL on treatment outcomes. However, there was insufficient data regarding adverse effects associated with the use of diuretics for further analysis. Differences in stone size, location, and type, as well as in the SWL machine used may have caused unpredictable bias. Moreover, unpublished and missing negative data may have resulted in bias toward the diuretic effects.

In conclusion, the use of diuretics significantly improves stone fragmentation in patients undergoing SWL. However, the improvement in stone clearance is insignificant.

## Supporting information

**S1 Checklist. Prisma Checklist for Effects of Diuretic Administration on Outcomes of Extracorporeal Shockwave Lithotripsy: A Systematic Review and Meta-analysis.** (DOC)

## Acknowledgments

The authors thank Xing-ming Zhang for his guidance on statistics.

## Author Contributions

**Conceptualization:** Zhenghao Wang, Yunjin Bai.

**Data curation:** Zhenghao Wang.

**Formal analysis:** Zhenghao Wang, Yunjin Bai.

**Funding acquisition:** Zhenghao Wang.

**Investigation:** Zhenghao Wang.

**Methodology:** Zhenghao Wang.

**Project administration:** Zhenghao Wang.

**Resources:** Zhenghao Wang.

**Software:** Zhenghao Wang, Yunjin Bai.

**Supervision:** Jia Wang.

**Validation:** Zhenghao Wang.

**Visualization:** Zhenghao Wang.

**Writing – original draft:** Zhenghao Wang.

**Writing – review & editing:** Zhenghao Wang.

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
