## [Decision Letter · Decision Letter 0]

14 Jan 2020

PONE-D-19-33308

Effects of DiureticAdministration on Outcomes of Extracorporeal Shockwave Lithotripsy: A Systematic Review and Meta-analysis

PLOS ONE

Dear Dr Wang,

Thank you for submitting your manuscript to PLOS ONE. After careful consideration, we feel that it has merit but does not fully meet PLOS ONE’s publication criteria as it currently stands. Therefore, we invite you to submit a revised version of the manuscript that addresses the points raised during the review process.

We would appreciate receiving your revised manuscript by 30 days. To enhance the reproducibility of your results, we recommend that if applicable you deposit your laboratory protocols in protocols.io, where a protocol can be assigned its own identifier (DOI) such that it can be cited independently in the future. For instructions see: http://journals.plos.org/plosone/s/submission-guidelines#loc-laboratory-protocols

We look forward to receiving your revised manuscript.

Kind regards,

Federico Bilotta

Academic Editor

PLOS ONE

Journal Requirements:

3. To comply with the items on the PRISMA checklist, please provide the following information in your Methods section: who screened studies for inclusion (which is a different step to data extraction). Please also structure your abstract using subheadings

Additional Editor Comments (if provided):

PONE-D-19-33308

In this SR and meta-analysis, the Authors investigated the effect of diuretics on the outcomes of shockwave lithotripsy (SWL) for the treatment of urinary stones.

The Authors performed searches of PubMed, Web of science, Embase, EBSCO, and Cochrane library databases from inception to November 2019.

The RCTs were selected for assessing the effects of diuretics on fragmentation and clearance of urinary stones. The search strategy and study selection process were performed in accordance with the PRISMA guidelines. Four RCTs were included in the meta-analysis.

Overall, intervention groups experienced significant improvements in fragmentation compared with the control groups (risk ratio [RR] = 1.14, 95% confidence interval [CI] = 1.05–1.03, P = 0.02).

However, stone clearance did not significantly differ between the intervention and control groups (RR = 1.23, 95% CI = 0.97–1.56, P = 0.08). The total numbers of shocks and sessions required were significantly reduced by the use of diuretics.

The Authors concluded that, diuretics significantly enhance stone fragmentation for patients undergoing SWL.

Reviewer comments: The article has analysed the available literature and extrapolated conclusions which are clinically insignificant to make recommendations to clinical practice or add any substantial value to the available literature. Reject

Editor’s comments: this SR and meta-analysis provide new information related the effect of diuretics on the outcomes of shockwave lithotripsy (SWL) for the treatment of urinary stones, despite some methodological limitations.

The Authors selected only 4 study to include in the meta-analysis.

The Table are indented in the text, please report them at the end of the article.

The first sentence of the Discussion should summarize the main results of the SR, please edit the sentence: “Following its implementation, SWL has since become the most frequently used approach for the treatment of urolithiasis (18).”

The Discussion section is long and straight, please focus on the outcome of the study and shortened by 20%.

I would like, for your interest, to remind one of paper related to the same topic that have been published in PLOS. Please consider to include:

1. Hai Wang, Libo Man, Guizhong Li, Guanglin Huang, Ning Liu, Jianwei Wang. Meta-Analysis of Stenting versus Non-Stenting for the Treatment of Ureteral Stones Research Article | published 09 Jan 2017 PLOS ONE. https://doi.org/10.1371/journal.pone.0167670

Reviewers' comments:

Reviewer's Responses to Questions

**Comments to the Author**

1. Is the manuscript technically sound, and do the data support the conclusions?

Reviewer #1: Partly

2. Has the statistical analysis been performed appropriately and rigorously? 

Reviewer #1: Yes

3. Have the authors made all data underlying the findings in their manuscript fully available?

Reviewer #1: Yes

4. Is the manuscript presented in an intelligible fashion and written in standard English?

Reviewer #1: Yes

5. Review Comments to the Author

Reviewer #1: The article has analysed the available litreature and extrapolated conclusions which are clinically insignificant to make recoomendations to clinical practice or add any substantial value to the available litreature.

6. PLOS authors have the option to publish the peer review history of their article (what does this mean?). If published, this will include your full peer review and any attached files.

Reviewer #1: No

---

## [Author Response · Author response to Decision Letter 0]

30 Jan 2020

1. The Table are indented in the text, please report them at the end of the article.

Response: We are very sorry to make the incorrect format and we corrected it according to requirement.

2. The first sentence of the Discussion should summarize the main results of the SR, please edit the sentence: “Following its implementation, SWL has since become the most frequently used approach for the treatment of urolithiasis (18).”

Response: It is a really good suggestion and we made the correction in this part,

3. The Discussion section is long and straight, please focus on the outcome of the study and shortened by 20%.

Response: Thanks for the comments and we have shortened this part according to the comments.

4. The paper related to the same topic that have been published in PLOS. Please consider to include: 1. Hai Wang, Libo Man, Guizhong Li, Guanglin Huang, Ning Liu, Jianwei Wang. Meta-Analysis of Stenting versus Non-Stenting for the Treatment of Ureteral Stones Research Article | published 09 Jan 2017 PLOS ONE. 

Response: This is a very impressive study and we have cited this study in our article as the suggestion.

---

## [Editor Report · Decision Letter 1]

21 Feb 2020

Effects of Diuretic Administration on Outcomes of Extracorporeal Shockwave Lithotripsy: A Systematic Review and Meta-analysis

PONE-D-19-33308R1

Dear Dr. Wang,

We are pleased to inform you that your manuscript has been judged scientifically suitable for publication and will be formally accepted for publication once it complies with all outstanding technical requirements.

With kind regards,

Federico Bilotta

Academic Editor

PLOS ONE
---

## [Editor Report · Acceptance letter]

27 Feb 2020

PONE-D-19-33308R1 

Effects of Diuretic Administration on Outcomes of Extracorporeal Shockwave Lithotripsy: A Systematic Review and Meta-analysis 

Dear Dr. Wang:

I am pleased to inform you that your manuscript has been deemed suitable for publication in PLOS ONE. Congratulations! Your manuscript is now with our production department. 

With kind regards,

on behalf of

Dr. Federico Bilotta 

Academic Editor

PLOS ONE